Identifying knowledge gaps in hypersaline systems supporting the global electrical transition: invertebrate community structure in salars from the lithium triangle

Salazar Gonzalo 1
Aguilar Pablo 2 3
Harrod Chris chris@harrodlab.net 4
1 Doctorado en Ciencias Aplicadas mención Sistemas Acuáticos, Facultad de Ciencias del Mar y Recursos Biológicos, Universidad de Antofagasta , Antofagasta , Chile
2 Departamento de Biotecnología, Facultad de Ciencias del Mar y de Recursos Biológicos, Universidad de Antofagasta , Antofagasta , Chile
3 Laboratorio de Complejidad Microbiana, Instituto Antofagasta, Universidad de Antofagasta , Antofagasta , Chile
4 Scottish Centre for Ecology and the Natural Environment, School of Biodiversity, One Health and Veterinary Medicine, University of Glasgow , Glasgow , United Kingdom
Brygadyrenko Viktor
Electronic publication date: 2025 Oct 13
Publication date: 2025
Volume: 13
Electronic Location ID: e20042
Received 2025 May 15; Accepted 2025 Aug 15
Copyright: ©2025 Salazar et al.
Copyright year: 2025
Copyright holder: Salazar et al.
License: This is an open access article distributed under the terms of the Creative Commons Attribution License, which permits unrestricted use, distribution, reproduction and adaptation in any medium and for any purpose provided that it is properly attributed. For attribution, the original author(s), title, publication source (PeerJ) and either DOI or URL of the article must be cited.
License URL: https://creativecommons.org/licenses/by/4.0/

Keywords: Atacama Desert, Altiplano, Benthos, Hypersaline lakes, Multivariate analysis, Zooplankton

Funding: ANID Beca Doctorado Nacional 1231904 EDGES Project (Curtin University) PLAN DE FORTALECIMIENTO UNIVERSIDADES ESTATALES- MINEDUC-CHILE RED21992 Proyecto Anillo, ANID ATE240021 This work was funded by several sources. GS was funded by ANID Beca Doctorado Nacional No. 1231904 and the EDGES Project (Curtin University). Support was also provided by PLAN DE FORTALECIMIENTO UNIVERSIDADES ESTATALES- MINEDUC-CHILE—RED21992. GS and PA were funded by Proyecto Anillo, ANID, ATE240021. The funders had no role in study design, data collection and analysis, decision to publish, or preparation of the manuscript.

==============================
Following decades of mining impacts, South America’s hypersaline wetlands (salars) face increasing threats from lithium extraction to support global decarbonisation. Although globally important, salars are understudied and information needed to understand environmental impacts is lacking. Modern ecological studies on salars have focused on microbial community composition and function but other taxa are less studied, including resident and migratory reptiles and birds and their aquatic invertebrate prey.

Given the scale and immediate nature of the threats associated with lithium exploitation, we must deepen our understanding of salar biology, but this is impeded by logistic/financial restrictions given the heightened costs of sampling in these often remote, extreme environments. Given the pressing demand for information, we collated/analysed information from the literature. We generated lists of invertebrate taxa present in 63 hypersaline environments from Argentina, Bolivia, Chile and Peru, and examined relationships between invertebrate community structure and physicochemical factors. We recorded 46 different taxa, with the Centropagidae being the most frequently recorded taxon across systems. Multivariate analyses of community structure showed significant clustering among sites. Variation in community structure was best explained by maximum salinity (18%). Geographical location or ecosystem size had no obvious effect on community structure. We provide a useful broad view of aquatic invertebrate diversity in the hypersaline salars but highlight the general lack of information regarding the ecology of these ecosystems.

Introduction

Salars—hypersaline ecosystems characterized by lakes, lagoons, wetlands, or a combination—are found in Argentina, Bolivia, Chile, and Peru (Risacher, Alonso & Salazar, 2003). Salars are usually found in endorheic basins associated with palaeo-lakes and have salinities (>40 g/L) that can far exceed those of oceanic waters (Gutiérrez et al., 2022). One of their most outstanding characteristics, in addition to their salt concentrations, is their elemental brine composition, which can include large amounts of lithium and boron (Godfrey et al., 2013; Álvarez-Amado et al., 2022). Although both elements are of industrial concern, lithium is currently of elevated global interest reflecting the high demand from the energy storage and automobile industries as a key tool to counter climate change. Fifty percent of lithium brine reserves are concentrated in Argentina, Bolivia, and Chile, forming an area known as the lithium triangle (USGS, 2024). Although Peru is not included in this area, it has salars, with interest recently expressed in exploiting lithium in Puno, in the SE of the country (Mares, 2022). Lithium exploitation in Argentina is currently carried out in two main operations, in the Salar del Hombre Muerto and in the Salar de Olaroz. Bolivian lithium production is currently limited to the Salar de Uyuni. In Chile, lithium production is focussed on the Salar de Atacama, Chile largest salar. However, recent realisation of the value of the resources have led to restructuring of the concessions regarding exploitation between 2016 and 2018 (Cabrera-Valencia, 2023), and the Chilean national government recently published a national lithium strategy (Gobierno de Chile, 2023), which includes a list of 7 salars (Salar de Atacama, Salar de Maricunga, Salar de Pedernales, Salar Grande, Salar de Infieles, Salar de La Isla and Salar de Aguilar) earmarked for lithium exploitation and other 6 (Salar de Coipasa, Salar de Ascotán, Salar de Ollagüe, Laguna Verde, Salar de Agua Amarga and Salar de Piedra Parada) under possible exploitation.

Thus, many of those inland hypersaline ecosystems face marked and increasing threats to their conservation and long-term existence due to direct factors such as extraction of water and other materials, e.g., minerals, and indirect factors including climate change, which individually, and in concert accelerate the drying process of these aquatic ecosystems (Gajardo & Redón, 2019; Gutiérrez et al., 2022). Mineral and water extraction affects the physicochemical properties of lakes, triggering changes in their biogeochemical attributes, thus affecting the often-specialised taxa that inhabit them (Acosta & Custodio, 2008; Ribera, 2016), and their capacity to provide ecosystem goods and services to human populations. The degradation of hypersaline lakes has occurred across their global distribution (Wurtsbaugh et al., 2017), but is a particular issue in the arid lithium triangle (Rentier, Hoorn & Seijmonsbergen, 2024).

Saline lakes provide various ecosystem services, including provision of water (for industrial mining, agriculture and municipal usage), provision of organisms or compounds for biotechnology and aquaculture, recreation, tourism, nature conservation and cultural services (Gajardo & Redón, 2019). Hypersaline lakes have also provided sensitive records of ecological, evolutionary and geological shifts through the formation and long-term retention of evaporites, which usually form in these ecosystems as a result of environmental processes (Oehlert et al., 2022). They also represent important dissolved inorganic carbon reservoirs on the planet, capturing atmospheric CO2 (Duarte et al., 2008).

Given the increasing threats to their conservation, there is a pressing need to deepen our understanding of the ecology and functioning of salars. As such, it is necessary to gather information on the complex ecological processes that occur in these systems to understand interactions between a broad (and understudied) group of taxa including microorganisms, vegetation, invertebrates and vertebrates. Most modern ecological studies conducted in salars have focused on the structure and function of the microbial community. Beyond revealing the existence of many extremophile taxa with adaptations to extreme conditions and unusual metabolisms, microbes have been shown to play major roles in fixing organic and inorganic compounds (e.g., carbon dioxide) and being key in the biogeochemical interactions of hypersaline lakes (Oren, 2011). However, salars support a diversity of other taxa, including aquatic invertebrates involved in key ecological processes in hypersaline ecosystems, such as the regulation of the proliferation of micro-organisms, which in turn regulates water turbidity and thus light penetration and stratification (Wurtsbaugh & Berry, 1990; Barnes & Wurtsbaugh, 2015). These organisms in turn are an important part of the diet of migratory and resident vertebrates such as birds that feed on micro-crustacea such as Artemia and insects e.g., Ephydridae larvae (Baxter, 2018). Looking more locally, there is no collective view of what aquatic invertebrate taxa are present in the salars of the lithium triangle, how they vary among systems, and if so, what factors are likely driving this variation.

There is a growing number of proposed projects to extract lithium from salars across the region to support global decarbonisation (Voskoboynik & Andreucci, 2022). Environmental Impact Assessments associated with such projects require reliable ecological information to support informed decision making. There is some information available on salar invertebrates from the lithium triangle, but most of it is dated (≥20 years old) and is potentially not relevant after several decades of over-exploitation of water and climate change. Furthermore, to our understanding there have been no efforts to use a community-based approach across salars, and approach needed to support ecosystem-based management approaches. There is a common perception among many stakeholders that these unique ecosystems are similar to the point that information from one is likely relevant for the management of all. However, over 30 years of work have shown that salars are not only widely different in regard to their physiochemical compositions but also their microbial ecology (Aguilar et al., 2016). It is likely that this extends to their macroinvertebrate community and their community structure and function. Although understudied in salar ecosystems, invertebrates from inland waters are useful indicators of the state of ecosystems, as they tend to be sensitive to anthropogenic changes (Collier, Probert & Jeffries, 2016). Benthic macroinvertebrates are widely used as bioindicators to assess riverine (Hawkes, 1998) or lacustrine water quality (Lindegaard, 1995). Aquatic invertebrates thus also have the potential to reflect the relative ecological status of salar ecosystems.

Given the current lack of information regarding the diversity and ecology of salar aquatic invertebrates we ask the following questions: What taxa are present in the salars of the lithium triangle? Is community structure similar across different salars? If not, are any physico-chemical factors associated with the presence or absence of some taxa? We undertook a desk-based review to compile data on aquatic invertebrates from those salars from those countries in South America potentially threatened by the lithium industry in order to address the questions presented above.

Material and Methods

Data were collated from studies conducted in Argentina, Bolivia, Chile and Peru by means of literature searches using Web of Science (https://www.webofscience.com), using the keywords “Saline Lakes” and the country of interest, as well as searching for the name of specific ecosystems from each of the four countries together with the keyword “invertebrates” (e.g., Salar de Atacama AND invertebrates). Searches were conducted in English and in Spanish. We also used the reference lists of these articles to encounter relevant studies.

This resulted in information gathered from 19 different articles and theses published between 1986 and 2022, detailing the presence of aquatic invertebrate taxa from a total of 63 sites (salt lakes, lagoons, salars: Fig. 1) from Argentina (n = 6), Bolivia (n = 27), Chile (n = 21) and Peru (n = 9). We generated a presence/absence matrix for invertebrate taxa recorded from these 19 articles. We also included data from two sites in Chile (Laguna Puilar and Salar de Tara) that we sampled by kick sampling in 2021. We also recorded relevant information where available including salar location, altitude above sea level, surface area, and maximum salinity. Where information was not provided in the article, we used Google Earth Pro to estimate altitude and lagoon surface area. If information on maximum salinity was not provided, we used data from Risacher & Fritz (1991) and Risacher, Alonso & Salazar (1999). It should be noted that not all articles took into account seasonal variability to study diversity at each sample site, and as such we did not consider this factor in our analyses.

Figure 1 Study area map depicting the location of the n = 63 salars and lagoons included in the current study.

The community matrix was based on the presence/absence of different aquatic invertebrate families due to the varying levels of taxonomic resolution reported across studies. We estimated family richness per salar and compared it to the different environmental variables using Pearson’s correlation. We then used a multivariate approach to examine patterns in aquatic invertebrate community structure across sites. First, a Jaccard similarity matrix was constructed. We then used group average hierarchical cluster analysis (where the new node takes the mean similarity of the individual nodes) to examine evidence for structuring within the dataset. Salars were placed into groups based on theirs proposed membership of different clusters. One-way permutational multivariate analysis of variance (PERMANOVA) tests (npermutations = 9999) were used to assess statistical support for differences in community structure between country and between putative clusters. A Similarity Percentages (SIMPER) test was used to highlight the taxa contributing to similarity within and dissimilarity between groups. An non-metric multidimensional scaling (nMDS) ordination was generated to visualize patterns in community structure between salars. Vectors were added to the multidimensional scaling (MDS) depicting the Pearson correlation between different taxa and the MDS axes, with the direction for each family indicating the sign and strength of the correlation. Finally, we used a distance based linear model (DistLM) to examine which environmental factors (altitude, maximum salinity, surface area, latitude, longitude) best explained variation in aquatic invertebrate community structure. Environmental data were normalised prior to their inclusion in the model as predictors. We used the BEST selection procedure which searches all possible combination of variables, and based our selection of best fit on the Akaike Information Criterion (AIC). All analyses were conducted in PRIMER/PERMANOVA+ 7.0.24 (Clarke & Gorley, 2015).

Figure 2 Variation in taxon frequency across the different sites included in the current study.

Results

There were more reports from Bolivian salars (n = 27), than Chile (n = 21), Peru (n = 9) or Argentina (n = 6). A total of 46 different aquatic invertebrate families were reported as being present in the 63 different salars (Tables 1, Table S1). The salars were distributed over a total area of 401 752 km2. The study area extended across 13 degrees of latitude and eight degrees of longitude. The most northerly salar was Parinacochas lagoon in Peru and the most southerly was Santa Rosa lagoon in Chile. Altitude varied between 750 and 4,690 m asl (mean ± SD = 3824 ± 823 m) and surface area ranged from <0.001 to 125 km2 (10.7 ± 23.8 km2). Maximum total salinity varied between 0.12 and 336 g/L.

Table 1 Data of the study sites considering the name of the salar or lagoon, the country, altitude, location, area and maximum salinity.

System name	Altitude	Longitude	Latitude	Country	Surface area (km2)	Maximum salinity (g/L)	N taxa	Reference	
Salar Pastos Grandes	4,440	−21.64	−67.80	Bolivia	125.00	5.70	6	Williams et al. (1995)	
Laguna Ramaditas	4,117	−21.63	−68.08	Bolivia	4.00	47.30	5	Dejoux (1993); Hurlbert, Lopez & Keith (1984); Williams et al. (1995)	
Laguna Hedionda	4,121	−21.57	−68.05	Bolivia	4.40	60.00	8	Dejoux (1993); Williams et al. (1995)	
Laguna Cañapa	4,140	−21.51	−68.01	Bolivia	1.50	85.80	5	Dejoux (1993); Williams et al. (1995)	
Laguna Colorada	4,278	−22.20	−67.78	Bolivia	52.00	156.40	2	Williams et al. (1995)	
Laguna Chiar Khota	4,110	−21.58	−68.07	Bolivia	2.10	120.00	1	Dejoux (1993)	
Laguna Honda	4,110	−21.62	−68.07	Bolivia	0.30	35.00	1	Dejoux (1993)	
Laguna Pujio	4,110	−21.62	−68.07	Bolivia	0.07	45.00	1	Dejoux (1993)	
Laguna Ballivian	4,130	−21.63	−68.08	Bolivia	0.02	36.00	2	Dejoux (1993)	
Laguna Caliente	4,440	−21.64	−67.80	Bolivia	0.34	38.00	8	Dejoux (1993)	
Laguna Cachi	4,495	−21.72	−67.95	Bolivia	1.10	12.40	3	Dejoux (1993)	
Polques	4,394	−22.53	−67.62	Bolivia	13.40	15.00	13	Bayly (1993); Dejoux (1993)	
Laguna Verde	4,310	−22.80	−67.80	Bolivia	16.00	58.10	7	Bayly (1993); Dejoux (1993)	
Aguas calientes I	4,280	−23.13	−67.40	Chile	2.50	122.89	1	Bayly (1993)	
Aguas calientes II	4,200	−23.52	−67.57	Chile	9.00	13.66	1	Bayly (1993)	
Aguas calientes III	3,950	−25.00	−68.63	Chile	2.50	25.15	1	Bayly (1993)	
Hombre muerto	3,973	−25.50	−66.85	Argentina	13.00	21.00	1	Bayly (1993)	
Guacha	4,406	−22.55	−67.52	Bolivia	2.10	36.00	1	Bayly (1993)	
Collpacocha	3,825	−15.25	−70.05	Peru	1.40	38.60	2	Hurlbert, Loayza & Moreno (1986); Bayly (1993)	
Este	4,407	−22.52	−67.48	Bolivia	0.52	86.00	1	Bayly (1993)	
Chojllas	4,545	−22.37	−67.10	Bolivia	5.50	11.10	1	Bayly (1993)	
Soledad	3,722	−17.73	−67.37	Bolivia	91.00	11.00	1	Bayly (1993)	
Loriscota	4,562	−16.87	−70.03	Peru	33.00	10.40	2	Bayly (1993); Hurlbert, Loayza & Moreno (1986)	
Khara	4,509	−21.90	−67.87	Bolivia	13.00	8.70	1	Bayly (1993)	
Puripica chico	4,393	−22.52	−67.50	Bolivia	0.70	8.20	1	Bayly (1993)	
Catalcito	4,545	−23.52	−67.25	Bolivia	2.10	8.10	2	Bayly (1993)	
Pozuelos	3,663	−22.33	−66.00	Argentina	87.80	6.20	1	Bayly (1993)	
Parinacochas	3,273	−15.28	−73.70	Peru	67.00	5.60	3	Bayly (1993); Hurlbert, Loayza & Moreno (1986)	
Pelada	4,590	−22.75	−67.17	Bolivia	1.90	4.10	2	Bayly (1993)	
Penitas blancas	4,530	−22.43	−67.37	Bolivia	0.10	3.70	1	Bayly (1993)	
Huancaroma	3,722	−17.67	−67.50	Bolivia	1.76	3.50	1	Bayly (1993)	
Campo Grande	4,553	−22.55	−67.20	Bolivia	2.10	3.40	1	Bayly (1993)	
Pampamarca	3,788	−14.13	−71.48	Peru	6.70	0.80	1	Bayly (1993)	
Viscacha	4,575	−16.88	−70.23	Peru	8.40	0.90	1	Bayly (1993); Hurlbert, Loayza & Moreno (1986)	
Conchostraca	4,690	−22.30	−67.23	Bolivia	0.13	0.69	1	Bayly (1993)	
Totoral	4,559	−22.53	−67.23	Bolivia	1.00	0.65	1	Bayly (1993)	
Loripongo	4,555	−16.83	−70.08	Peru	0.60	0.62	2	Bayly (1993); Hurlbert, Loayza & Moreno (1986)	
Llamara	754	−21.30	−69.62	Chile	0.02	160.00	1	De Los Ríos-Escalante (2005); Zúñiga et al. (1999)	
Cejas I	2,343	−23.03	−68.22	Chile	0.03	129.30	1	De Los Ríos-Escalante (2005); Zúñiga et al. (1999); Zúñiga et al. (1994)	
Cejas II	2,343	−23.03	−68.22	Chile	0.02	150.00	1	De los Ríos-Escalante & Amarouayache (2016)	
Cejas III	2,343	−23.03	−68.22	Chile	0.07	189.00	1	De los Ríos-Escalante & Amarouayache (2016)	
Tebenquiche	2,317	−23.12	−68.27	Chile	2.00	300.00	2	De Los Ríos-Escalante (2005); Zúñiga et al. (1999); Zúñiga et al. (1994)	
Chaxas	2,302	−23.29	−68.18	Chile	0.20	120.00	1	De Los Ríos-Escalante (2005)	
Gemela Este	2,400	−23.23	−68.23	Chile	1.00	41.00	1	De Los Ríos-Escalante (2005); De Los Ríos-Escalante, Fransen & Klein (2010)	
Gemela Oeste	2,400	−23.23	−68.23	Chile	1.00	51.00	1	De Los Ríos-Escalante (2005); De Los Ríos-Escalante, Fransen & Klein (2010)	
Miscanti	4,120	−23.72	−67.80	Chile	13.40	8.98	4	De Los Ríos-Escalante (2005); De Los Ríos-Escalante, Fransen & Klein (2010)	
Miñiques	4,120	−23.72	−67.80	Chile	1.60	9.79	4	De Los Ríos-Escalante (2005); De Los Ríos-Escalante, Fransen & Klein (2010)	
Capur	3,950	−23.90	−67.80	Chile	0.90	3.40	3	De Los Ríos-Escalante (2005); De Los Ríos-Escalante, Fransen & Klein (2010)	
Santa Rosa	3,766	−27.08	−69.17	Chile	0.64	8.00	2	Bayly (1993); De Los Ríos-Escalante (2005); De Los Ríos-Escalante, Fransen & Klein (2010)	
Tilopozo	2,314	−23.78	−68.24	Chile	0.20	3.00	6	Collado, Valladares & Méndez (2013); Zúñiga et al. (1991)	
Puilar	2,306	−23.31	−68.15	Chile	0.12	16.98	6	(Dorador et al., 2018; G. Salazar, 2025, unpublished data)	
Tara	4,322	−23.02	−67.30	Chile	14.00	70.00	12	(García-Sanz et al., 2021; G. Salazar, 2025, unpublished data)	
Ascotan	3,716	−21.49	−68.26	Chile	18.00	119.85	21	Collado & Méndez (2012); Lagomarsino-Pizarro (2016); Sobarzo-Opazo (2014)	
Carcote	3,690	−21.27	−68.32	Chile	3.50	335.54	11	Cárcamo-Téjer (2017); Collado & Méndez (2013)	
Laguna Verde (Argentina)	3334	−25.48	−67.55	Argentina	0.01	176.26	5	Colla, Lencina & Farías (2022)	
Pozo Bravo	3,327	−25.52	−67.58	Argentina	0.02	140.69	5	Colla, Lencina & Farías (2022)	
Ojo de campo azul	3,331	−25.61	−67.67	Argentina	0.002	27.90	11	Colla, Lencina & Farías (2022)	
Ojo de campo naranja	3,332	−25.61	−67.67	Argentina	0.001	149.60	2	Colla, Lencina & Farías (2022)	
Salar de Surire	4,260	−18.84	−69.05	Chile	9.50	102.00	16	Scheihing et al. (2010); Zúñiga et al. (1999)	
Lago Salinas	3,840	−14.98	−70.12	Peru	9.70	251.00	1	Hurlbert, Loayza & Moreno (1986)	
Saytococha	4,225	−15.90	−70.53	Peru	1.00	0.12	5	Hurlbert, Loayza & Moreno (1986)	
Laguna las Salinas	4,295	−16.37	−71.15	Peru	24.00	8.50	2	Hurlbert, Loayza & Moreno (1986)	
Laguna Chulluncani	4,450	−21.55	−67.88	Bolivia	0.80	69.00	2	Hurlbert, Lopez & Keith (1984)	

The number of families reported per salar varied between one and 21 families with a mean (±SD) of 3.4 (±4.0). There was no apparent relationship (Pearson r = 0.04 to 0.16, P = 0.21 to 0.85) between family richness reported from a given salar and any of the different environmental variables (i.e., altitude, surface area, maximum salinity, longitude or latitude.

Some patterns that can be observed in the data are that as salinity increases the number of taxa is mostly reduced, with Artemiidae being the dominant group under these conditions. Also, the copepod family Centropagidae (present in 39 sites) was the most frequent group present across sites, represented by the genus Boeckella (Fig. 2).

A one-way PERMANOVA provided some evidence (PseudoF3,59 = 2.17, P = 0.009) that aquatic invertebrate community structure differed in the salars in the four countries. Pairwise comparisons showed that only in Bolivia and Chile (t-test: P = 0.02) and Argentina and Peru (P = 0.04) were differences in community structure significantly different. Group average hierarchical cluster analysis provided strong evidence that the aquatic invertebrate community structure differed between individual salars and that they could be reliably clustered into eight groups (Fig. 3). A PERMANOVA analysis provided strong statistical support for the cluster analysis (Psuedo-F7,55 = 22.12, P = 0.001). The geographical location of salars belonging to the different clusters was not based on political boundaries (Fig. 3). The nMDS ordination (Fig. 4) provided more support for the separation of the salars into groups (Stress 0.06). The first MDS axis was positively associated with the Centropagidae and negatively with the Artemiidae (Fig. 4), while the second MDS axis was positively associated with a range of different families.

Figure 3 Dendogram (Ward’s cluster analysis based on a Jaccard similarity matrix) showing variation in aquatic invertebrate community structure across n = 63 salars and lagoons.

There was no obvious geographical pattern in the distribution of the different clusters.

Figure 4 NMDS ordination depicting variation in aquatic invertebrate community structure from the salars and lagoons included in the current study.

Marker colour reflects cluster membership (see Fig. 3). Vectors reflect the relative strength and direction of correlations between the presence of certain taxa and ordination scores.

The results of the SIMPER test allowed a deeper understanding of the invertebrate families associated with the different clusters. Group A was characterised by Centropagidae and Cyclopidae (both 37%); Group B by Centropagidae (99%); Group C, Artemiidae (98%); Group E, Hyalellidae (30%) and Ephydridae (21%); Group F, Hyalellidae (21%) and Chironomidae (12%); Group G, Chironomidae (100%); Group H, Chironomidae (33%) and Ephydridae (33%). Group D was based on a single site so was not considered for the analysis but was most similar to Group F.

The DistLM test indicated that the best model (AIC = 507) accounting for the variation in aquatic invertebrate community structure across the 63 different salars included in our analysis was based on two variables: maximum salinity and altitude, which combined explained 18.7% of the variation (Fig. 5). The bulk of the exploratory power was associated with maximum salinity (15.9%).

Figure 5 Distance based redundancy analysis showing environmental factors selected by the model and the percentage of variation associated with each factor.

Discussion

Reflecting the ongoing threats to salar ecosystems across South America, and the limited understanding of key components of their ecology, this study used a literature-based approach to examine several questions concerning the aquatic invertebrate communities found in salars and lagoons throughout the lithium triangle.

Regarding our first question (what aquatic invertebrate taxa were present in the salars of the lithium triangle) we provided information that the taxa present are depauperate: only 46 families of aquatic invertebrates were recorded from 63 salars and lagoons. Our second question (is community structure similar across different sites?) was answered in the negative, with clear evidence of clustering across the different sites. Several families dominated records including the Centropagidae followed by the Chironomidae, Artemiidae, Cyclopidae, Ephydridae, Hyalellidae, Canthocamptidae and Elmidae. In terms of their likely role in the ecosystem, zooplanktonic organisms such as Artemiidae, Centropagidae and Cyclopidae consume microalgae and microorganisms in the water column, that affect water turbidity and depth light penetration (Saccò et al., 2021). In the case of benthic organisms such as the larvae of Chironomidae, Ephydridae, Hyalellidae, Canthocamptidae and Elmidae, they play a crucial role in the degradation of organic matter present on the bottom (Covich, Palmer & Crowl, 1999; Schratzberger & Ingels, 2018), which can be used by smaller organisms such as bacteria, and in turn these invertebrates represent a food source for migratory or resident birds in these ecosystems. Artemiidae and Ephydridae were associated with higher salinities, and can be found in systems with salinities >200 g/L (Herbst, Conte & Brookes, 1988; De Vos et al., 2019) respectively. Some taxa, such as the Centropagidae represented by the species Boeckella poopoensis, are able to tolerate salinities up to 90 g/L (De Los Rios & Crespo, 2004). An interesting aspect of this species is their capacity to vary their reproductive strategy throughout the hydroperiod, thriving when salinity increases and preventing the presence of less tolerant species (Vignatti, Cecilia & Echaniz, 2016). However, the other species in this family tend to be found at much lower salinities of around 1 g/L (De Los Ríos-Escalante & Contreras, 2005). Chironomidae have been found at salinities of up to 150–170 g/L, with high abundances reported from Crimea, although several species are usually found at lower salinities (Belyakov et al., 2018). The family Canthocamptidae is a very poorly studied family with respect to the habitats of the lithium triangle, the only species described being Cletocamptus cecsurirensis from the Salar de Surire in Chile (Gómez, Scheihing & Labarca, 2007). However, as a group the harpacticoid copepods have representatives capable of tolerating high salinities (100 g/L) as is the case of Tigriopus brevicornis (McAllen, Taylor & Davenport, 1998). Hyalellidae and Elmidae are usually found at lower salinities, with both families being associated with fresh and brackish water (Domínguez & Fernández, 2009).

Although we only included a limited set of environmental variables, regarding the third question of this study, we found that there were indeed factors that were associated with the community structure variation of aquatic invertebrates in the salars, the most important factor being salinity, due to the stronger association (ca. 15%) shown by the distance-based linear model. Altitude contributed significantly but provided far less (∽3%) explanatory power. The role of maximum salinity reflects its importance as a key driver of aquatic invertebrate community structure at the global level (Anufriieva & Shadrin, 2018; Saccò et al., 2021).

Salar area, latitude or longitude had no measurable influence on macroinvertebrate community structure. This is quite striking, as it has been reported that species richness tends to increase with increasing lake area (Arnott, 2009). However, it is also relevant to note that this effect of species richness in relation to area has a direct effect with the size of the study organism, with this effect being more marked in larger aquatic organisms (Dawson et al., 2016). Conversely marine lakes, which are isolated systems that are fed by seawater through crevices and rainwater, have shown a direct relationship with lake area and species richness. Although the salars of the South American Altiplano are isolated on the surface, some basins have underground water connections reflecting their previous common origin as paleo-lakes (Risacher, Alonso & Salazar, 2003; Pfeiffer et al., 2018). The inclusion of zooplanktonic taxa may also have affected our results given their superior dispersal capacity relative to other invertebrate taxa (Arnott, 2009). In addition to this, there are also relevant factors to consider including temporal variation within sites. For example, zooplankton community structure can vary across seasons (Arnott et al., 1999). Given the limited information available, we could not include within-system variation but there is a clear need to examine how invertebrate community composition varies within different salars in order to allow fully robust comparisons. Our data show that although the salars include some common taxa, they are individually ecologically distinct. These differences are likely to be due to differences in the physicochemical and chemical characteristics present but also reflect differences in the functioning of the salars and lagoons. We showed that salars and lagoons belonging to similar geographic locations belonged to different clusters. Conversely, salars belonging to similar clusters were often found in quite different areas across the >400,000 km2 study area. These results highlight the importance of considering each salar on its own merits when considering its management, conservation or exploitation. Subject to the differences in sampling methods and reporting approach described above, it is quite likely that different salars in the same general geographic area will have markedly different invertebrate community structures.

Other salt lakes in the world, such as those found in Tibet, share some characteristics with South American salars, such as being at high altitude (>4,000 m asl) and having lithium-rich brines which are currently being exploited (Ding et al., 2023). Articles describing the diversity of aquatic invertebrates from Tibetan salt lakes, have highlighted Artemia species as the dominant group of invertebrates in hypersaline conditions. Other halotolerant (≥200 g/L) invertebrates recorded from Tibetan salt lakes include Eucypris inflata, Keratella quadrata, Brachionus plicatilis and Ephydra, with the highest diversity being recorded in brackish and moderate salinities with representatives of Rotifera (Brachionus, Conquiloides, Keratella, Lecane, Lepadella, Notholca, Polyarthra and Philodina), Copepoda (Cletocamptus, Arctodiaptomus and Eucyclops), Cladocera (Daphnia, Daphniopsis, Alona and Chydorus), Ostracoda (Cypris), Nematoda, Amphipoda (Gammarus sp.) and Mollusca (Radix and Hippeutis) (Williams, 1991; Mianping et al., 1993; Wen et al., 2005). Studies from Tibet include greater taxonomic resolution compared to the work conducted and summarised here from the lithium triangle, but some taxa have a certain similarity across the two distinct geographical areas. Conversely, there is little available information regarding the biodiversity impacts of lithium brine extraction in the Tibetan salt lakes, however, it is expected that these will be similar to those already mentioned for the South American salars.

Despite finding marked variation in aquatic invertebrate community structure in salar systems located throughout Argentina, Bolivia, Chile and Peru, it is important to note some limitations that potentially affect our results and conclusions. First, our data largely come from different published studies, generated by different researchers, over different years (from the 1980s to the present), with different methods and different questions (study of a single group of taxa or study of the entire invertebrate community). However, these data and our analyses are useful and have revealed key gaps in our understanding.

Future studies of salar ecology need to use standardised monitoring methods to estimate taxonomy and abundance/biomass in invertebrates and need to include the statistical power required to identify trends in abundance. They also need to include seasonal variation, which is likely crucial in determining invertebrate community structure, especially for zooplankton. This study has focused on studies using traditional morphological identification of salar taxa, but there is clear scope to use molecular approaches such as eDNA metabarcoding in conjunction with these traditional methods. The use of eDNA in salar ecosystems has great potential, but a lack of local taxa in DNA libraries results in misidentification and the approach may even not record taxa that are present (Saccò et al., 2025). Before eDNA will be fully useful in the study of salar ecosystems, there is a need to develop updated and local taxonomic keys to identify the species present in salars as well as sampling and sequencing of salar taxa across the four countries.

This study has highlighted how little we know about the diversity and ecology of aquatic invertebrates in the salars of the lithium triangle. A key knowledge gap exists regarding the sensitivity of these taxa to future physico-chemical changes in the salars. Published information on the impacts of direct extraction of brine from the salars in conjunction with climate change will likely bring irreversible changes. Open water areas will shrink, and salinities will increase (Gajardo & Redón, 2019) to levels that many invertebrate taxa will not be able to tolerate. Given that South American salars do not have a high diversity of halotolerant invertebrates compared to other hypersaline lakes like Australia (Lawrie, Chaplin & Pinder, 2021), this could be a clear sign that those invertebrates that are associated with freshwater and brackish waters (which comprises the majority of the taxa recorded for this study) will be the first to disappear as salinity increases due to indiscriminate lithium mining.

Understanding that salars support different invertebrate communities across the wide diversity of salars present, it is important that developers do not consider all salars to be the same, especially because aquatic invertebrates are a group of metazoans sensitive to anthropogenic disturbances. Considering the functional importance of invertebrates in biogeochemical cycles, and their role as a key food source for consumers including birds of critical conservation status, they are a valuable object of study in a scenario of ecosystem loss due to the direct action of man in water extraction and climate change.

Conclusion

This first region-wide analysis of salar aquatic invertebrate community structure across the lithium triangle shows that taxon richness and community structure vary considerably across sites. Salars cannot be regarded as homogeneous and will need to be considered individually (i.e., during environmental impact assessments associated with lithium exploitation). Variation in community structure was not associated with location within the region or surface area, and appears to be largely driven by maximum salinity (and to a lesser degree, altitude). Artemiidae are typically assumed to dominate saline lakes in the region, but results reveal that Centropagidae are most common taxon recorded. Despite these limitations, this study provides a useful analysis of the information available and opens many areas for future research. Developers must assess each salar individually prior to lithium exploitation.

Supplemental Information

Supplemental Information 1 Salar environmental and taxonomic data used for the analysis

We thank the editor and three anonymous reviewers for their time and useful comments.

Additional Information and Declarations

Competing Interests

Author Contributions

Data Availability

The authors declare there are no competing interests.

Gonzalo Salazar conceived and designed the experiments, performed the experiments, analyzed the data, prepared figures and/or tables, authored or reviewed drafts of the article, and approved the final draft.

Pablo Aguilar conceived and designed the experiments, performed the experiments, analyzed the data, authored or reviewed drafts of the article, and approved the final draft.

Chris Harrod conceived and designed the experiments, performed the experiments, analyzed the data, prepared figures and/or tables, authored or reviewed drafts of the article, and approved the final draft.

The following information was supplied regarding data availability:

The community structure data and environmental data that all analyses are based on are included in Supplementary File 1.

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
