# Peer review of "Identifying knowledge gaps in hypersaline systems supporting the global electrical transition: invertebrate community structure in salars from the lithium triangle"

_PeerJ, doi:10.7717/peerj.20042_

## Round 0.1 · original submission · Major Revisions

Dear Dr. Harrod, I ask you to carefully supplement the manuscript in accordance with the reviewers' comments. I hope that the new version of this article will be recommended for publication by the reviewers.

Reviewer 1 ·

Basic reporting

The research topic is highly relevant and compelling. It highlights the antagonism between the potential of new technologies to preserve global ecosystems at a broad scale and the need to extract mineral resources, which poses a threat to specific ecosystems. Addressing this issue is an important challenge that goes far beyond a single case study. Hypersaline ecosystems exist under extreme environmental conditions. On one hand, they provide habitats for rare species; on the other hand, they are highly sensitive to environmental disturbances. This makes the research question particularly interesting. The authors provide a thorough and well-structured review of the literature and clearly justify the objectives of their study.

Experimental design

The authors conducted a meta-analysis of existing data from the scientific literature and compiled a faunistic database. This database was enriched with results from their own field surveys. For analyzing the species matrix, the authors applied well-established ecological methods and described the analytical protocol in detail. This section of the manuscript is satisfactory and meets the standards of scientific publication.

Validity of the findings

The results are presented clearly, concisely, and coherently. The findings are well illustrated, and the statistical procedures are appropriately applied.
The authors note that:

“There was no apparent relationship (Pearson r = 0.04 to 0.16, P = 0.21 to 0.85) between family richness reported from a given salar and any of the different environmental variables (i.e., altitude, surface area, maximum salinity, longitude or latitude).”

It would be interesting to explore a nonlinear response model of family richness to environmental variables, as such relationships are often expected in broad-scale ecological datasets.

The authors appropriately discuss the limitations of their approach and outline directions for future research, as well as the potential practical applications of their findings.

·

Basic reporting

The manuscript is devoted to the analysis of the biodiversity of invertebrate communities in various saline reservoirs of some countries of Latine America, where rare earth metals, in particular, lithium, are mined. The analysis is based on publications. Using various mathematical methods, the authors have identified the existence of possible links between certain physico-chemical characteristics ofsaline reservoirs and the composition of biota. In general, the study is of some interest and provides information about biota and its features in different saline reservoirs. However, the study is mainly descriptive in nature and the data presented are mostly well-known in thepublications, do not contain fundamentally new scientific information and may be of local interest. In addition, there are a number of problems with this manuscript.

Experimental design

Material and Methods
1. Have the seasons of the research been taken into account in the publications? The diversity and abundance of species in salt lakes has a pronounced seasonality.
2. Have you considered diversity indexes (for example, the Shannon Index, etc.)? They were very useful for the purposes of this study.

Validity of the findings

Abstract
The authors postulate, that “Modern ecological studies on salars have focused on microbial community composition and function but other taxa are less…” However, this is not the case, there are many publications on Artemia populations in salt lakes and their role in the ecosystem, aquaculture and etc. What specific environmental threats to salars are associated with lithium mining?
Introduction
p.44-46 - it is necessary to specify the salinity concentration in salinas (see Chao Liang et al., 2024)
p. 46- 47 One of their most outstanding characteristics, in addition to their salt concentrations, is their elemental composition, which can include large amounts of lithium and boron – references?
In what other salars of the world are lithium mined? It is necessary to add information about the salt lakes of Tibet (China)
Results
p.167-168 Maximum total salinity varied between 0.12 and 336. - What are the salinity units?
p. 174-175. “Some patterns that can be observed in the data are that as salinity increases the number of taxa is mostly reduced, with Artemiidae being the dominant group under these conditions” - This is a well-known information.

Additional comments

Discussion
There is no fundamentally new information about the ecology of salt lakes in the discussion. In the publications presented, there is no information about the seasons of the study, which significantly affect the biota in saline reservoirs. The discussion contains general phrases, there is no comparison with other geographical areas and saline reservoirs used in lithium mining (Tibet). What are the possible consequences for the ecology of these reservoirs during lithium mining? What is the most vulnerable taxa? What are the possible consequences of anthropogenic influence on these reservoirs in the case of lithium production? The work is descriptive in nature, rather artificial connections between different parameters have been established based on literary data obtained in different years and in different seasons. The study has a pronounced local importance and can be used in the organization of management on a specific reservoir under consideration.

---

## Round 0.2 · Minor Revisions

Dear Dr. Harrod, Before this article is approved by the reviewers, I ask you to correct a few more shortcomings.

Reviewer 1 ·

Basic reporting

The authors have implemented all the recommendations of the reviewer. The quality of the manuscript has been significantly improved. I recommend the article for publication.

Experimental design

The authors have implemented all the recommendations of the reviewer. The quality of the manuscript has been significantly improved. I recommend the article for publication.

Validity of the findings

The authors have implemented all the recommendations of the reviewer. The quality of the manuscript has been significantly improved. I recommend the article for publication.

Reviewer 3 ·

Basic reporting

The manuscript is written in professional and comprehensible English. The terminology is accurate, and the narrative flows logically. The introduction clearly contextualizes the study within the broader discussion of lithium extraction and its ecological consequences. The rationale for focusing on aquatic macroinvertebrates is convincingly presented. The references are current, relevant, and appropriately cited throughout. The manuscript follows PeerJ’s structural standards. Figures and tables are relevant, clearly labelled, and of high quality. Supplementary data files support transparency and reproducibility.

Experimental design

The study addresses a significant knowledge gap: the limited understanding of macroinvertebrate diversity and community structure in hypersaline ecosystems threatened by lithium extraction. The literature-based approach is justified, given the logistical challenges of field sampling.

Validity of the findings

The authors synthesize data from 63 salars based on 19 studies and two field sites. Despite heterogeneity in methods, the dataset is large and geographically broad. Statistical approaches are appropriate for the data type.

Additional comments

I recommend removing the words "Identifying" and "supporting the global electrical transition" from the title of the article. Perhaps it would be better to add "(South America)" in brackets at the end of the article. These are my recommendations, not a requirement. The title should be short, informative, and understandable.
For better visibility of the Abstract in the Scopus and Web of Science databases, I recommend adding the Latin names of 10-12 invertebrates. This way, specialists in these groups of invertebrates will see your article.
Unsuccessful keywords: "Atacama Desert, Altiplano, benthos, hypersaline lakes, multivariate analysis, zooplankton". It would be better to simply mention the first two geographical names in the text of the Abstract and remove them from the keywords. The phrase "multivariate analysis" is uninformative (a query in Web of Science will give tens of thousands of results that are not related to your article. Think about what other 7 phrases would be better to add to the keywords.
The rounding of numbers in Table 1 in the columns "Surface area (km2)" and "Maximum salinity (g/L)" should be the same within each of the columns.
Figure 4 is unsuccessful: it will take up a whole page, while half of the page is empty space and a legend. Think about how to make the image more compact.
There is no need to put the legend in Figures 3-5 in a black mourning frame. The legends in Figures 3 to 5 are enclosed in a thick black frame, which is not present in Figure 1 and may be visually distracting. For consistency and improved visual appeal, I recommend removing the black border around the figure legends. The simpler format used in Figure 1 is clearer and more aesthetically pleasing.
In Figure 4, several taxon labels appear to overlap within the nMDS ordination plot, making them difficult to read. I recommend adjusting the position of overlapping text or using callouts/offset labels to improve clarity. Clear labeling is essential for interpreting vector directions and taxon associations.
Within one bracket in the text of the article, sources should be arranged by year.

---

## Round 0.3 · accepted · Accept

Dear Dr. Harrod, I congratulate you on the acceptance of this article for publication.